

# Genetic population structure of the pelagic mollusk *Limacina helicina* in the Kara Sea

Galina Anatolievna Abyzova[1], Mikhail Aleksandrovich Nikitin[2], Olga Vladimirovna Popova[2] and Anna Fedorovna Pasternak[1]

[1] Shirshov Institute of Oceanology, Russian Academy of Sciences, Moscow, Russia
[2] Belozersky Institute for Physico-Chemical Biology, Lomonosov Moscow State University, Moscow, Russia

## ABSTRACT

**Background**. Pelagic pteropods *Limacina helicina* are widespread and can play an important role in the food webs and in biosedimentation in Arctic and Subarctic ecosystems. Previous publications have shown differences in the genetic structure of populations of *L. helicina* from populations found in the Pacific Ocean and Svalbard area. Currently, there are no data on the genetic structure of *L. helicina* populations in the seas of the Siberian Arctic. We assessed the genetic structure of *L. helicina* from the Kara Sea populations and compared them with samples from around Svalbard and the North Pacific.

**Methods**. We examined genetic differences in *L. helicina* from three different locations in the Kara Sea via analysis of a fragment of the mitochondrial gene COI. We also compared a subset of samples with *L. helicina* from previous studies to find connections between populations from the Atlantic and Pacific Oceans.

**Results**. 65 individual *L. helinica* from the Kara Sea were sequenced to produce 19 different haplotypes. This is comparable with numbers of haplotypes found in Svalbard and Pacific samples (24 and 25, respectively). Haplotypes from different locations sampled around the Arctic and Subarctic were combined into two different groups: H1 and H2. The H2 includes sequences from the Kara Sea and Svalbard, was present only in the Atlantic sector of the Arctic. The other genetic group, H1, is widespread and found throughout all *L. helicina* populations. $\phi$ ST analyses also indicated significant genetic difference between the Atlantic and Pacific regions, but no differences between Svalbard and the Kara Sea.

**Discussion**. The obtained results support our hypothesis about genetic similarity of *L. helicina* populations from the Kara Sea and Svalbard: the majority of haplotypes belongs to the haplotype group H2, with the H1 group representing a minority of the haplotypes present. In contrast, in the Canadian Arctic and the Pacific Ocean only haplogroup H1 is found. The negative values of Fu's Fs indicate directed selection or expansion of the population. The reason for this pattern could be an isolation of the *Limacina helicina* population during the Pleistocene glaciation and a subsequent rapid expansion of this species after the last glacial maximum.

Corresponding author
Galina Anatolievna Abyzova,
abyzova.ga@ocean.ru

## INTRODUCTION

Pelagic pteropods *Limacina helicina* (Phipps, 1774) are widespread in marine Arctic and Subarctic ecosystems, where their local abundance and biomass are comparable to or greater than that of copepods (*Bernard & Froneman, 2005*; *Hunt et al., 2008*). Pteropods are able to form locally dense aggregations in the water column (*Percy & Fife, 1985*). *Limacina helicina* is the main food of many zooplankton organisms and predators of higher trophic levels, such as fish, whales, and birds (*Hunt et al., 2008*), and play a key role in the food web and in biosedimentation (*Gilmer & Harbison, 1986*; *Noji et al., 1997*; *Bernard & Froneman, 2009*; *Manno et al., 2009*).

The body of *Limacina helicina* is covered by a fragile calcium carbonate shell that protects them from predation. The aragonite composition of the shell makes these animals extremely sensitive to ocean acidification, which is expected to increase due to anthropogenic $CO_2$ emissions into the atmosphere (*Teniswood et al., 2016*). Consequently, this species represents a good model organism for ecological, physiological and biogeographical studies on how climate change is affecting the Arctic Ocean (*Comeau et al., 2009*; *Lischka et al., 2011*).

Despite the important role of pteropods in Arctic ecosystems, little is known about the genetic structure of *L. helicina* populations. A high diversity of haplotypes was found in local populations from the fjords of Svalbard (*Sromek, Lasota & Wolowicz, 2015*), including haplotypes typical of these pteropods in the Pacific Ocean (*Shimizu et al., 2017*). However, studies on the genetic structure of *L. helicina* have not been carried out in the Siberian Arctic seas.

In the Kara Sea, pteropods are a common component of the pelagic community and their spatial distribution is patchy (*Arashkevich et al., 2010*; *Flint et al., 2015*) similar to the other areas (*Percy & Fife, 1985*). Within patches, their abundance reached one million ind. m$^{-2}$, and they are the dominant consumers of suspended matter and phytoplankton (*Drits et al., 2015*).

The Kara Sea is a typical shelf Siberian Arctic Sea the warm, salty water from the Barents Sea enters from the south into the Kara Sea, and the cold Arctic water penetrates from the north (*Zatsepin et al., 2015*). We expect that the genetic structure of the populations of *L. helicina* in the Kara Sea is similar to that in the Svalbard region, which is also influenced by the Barents Sea and Arctic basin waters (*Stiansen & Filin, 2007*). In the Kara Sea, however, the effect of waters of different origin, combined with the impact of a strong river run-off, creates a mosaic of biotopes, where the genetic structure of populations can be different. We tested these hypotheses by examining intraspecific diversity of *L. helicina* in the Kara Sea using a fragment of the mitochondrial gene COI.

## MATERIALS & METHODS

*L. helicina* were selected from zooplankton samples collected during the cruise #63 of the RV Akademik Mstislav Keldysh in the Kara Sea that took place September-October 2015. Samples were collected at three different locations: station 5265 in the south of the Kara Sea and two stations in the Voronin and St. Anna troughs, 5239 and 5212, respectively

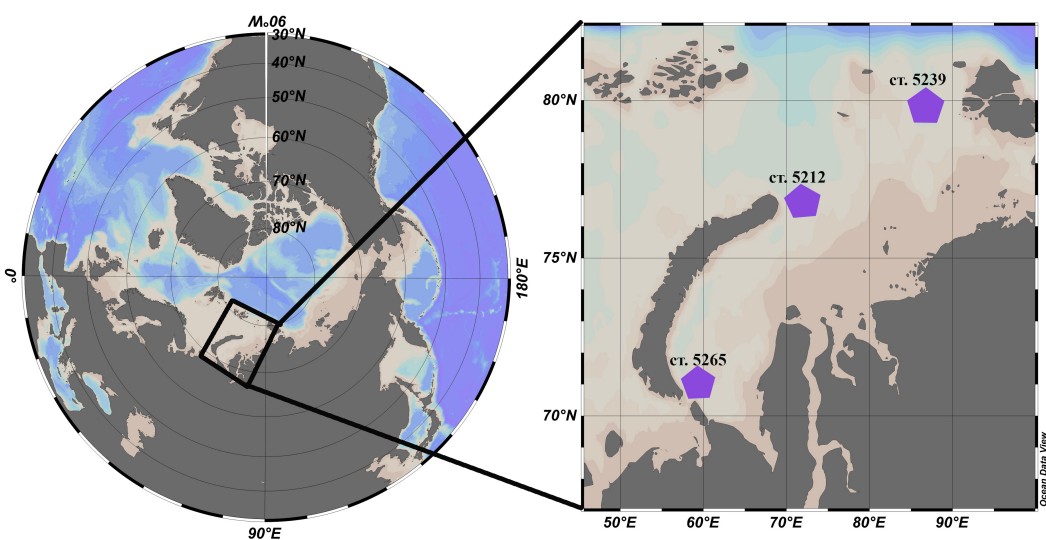

**Figure 1** Location of the stations in the Kara Sea where *L. helicina* were collected. *Schlitzer (2018)*.

(Fig. 1, Table 1). In the top 20 m of the water column at the station AMK 5265 temperature was 6 °C and salinity was 31.4. At station AMK 5239 temperature was 1.2 °C and salinity was 30 (this station is affected by freshwater runoff and melting ice), while at station AMK 5212 in the St. Anna trough was 4.3 °C and salinity was 34.3.

Pteropods were preserved in 96% ethanol immediately after collection. DNA was isolated from a piece of the pteropodia of large individuals (1–7 mm) or from the whole animal in case of small individuals (0.1–0.7 mm) using the ExtraGene™ DNA Prep 100 kit (Isogen, Moscow, Russia) per the manufacturer's protocol.

Fragments of mitochondrial cytochrome oxidase subunit gene (COI) were amplified using Encyclo Plus PCR kit (Eurogen, Moscow, Russia) using two standard primers: LCO-1490 (5′-GTCAACAAATCATAAAGATATTGG-3′) and HCO-2198 (5′-TAAACTTCAGGGTGACCAAAAAATCA-3′) (*Folmer et al., 1994*). PCR was conducted using the following common PCR cycle settings: 5 min at 95 °C, 40 cycles of 95 °C for 30 s, followed by annealing at 48 °C for 45 s, 72 °C for 1 min, and then a final elongation at 72 °C for 5 min. PCR products were analyzed with a 1% agarose gel electrophoresis, purified and sequenced using an Applied Biosystems® 3500 Genetic Analyzer. Subsequently sequences were aligned and analyzed using MEGA 6.0 (*Tamura et al., 2013*). The 503 bp fragments of COI gene were used for comparison with all other *L. helicina* samples from the Arctic and Pacific available from the GenBank database (Table 1). Low quality contigs (contigs containing more than 3 Ns) were excluded from analysis. The software Popart 1.7 (*Leigh & Bryant, 2015*) was then used for comparative analysis and identification of differences between populations as well as for construction of a TCS haplotype network (*Clement et al., 2002*). Furthermore, the program DnaSP (*Rozas et al., 2017*) was used for an estimation of genetic diversity in populations. Finally, the Arlequin 3.5 (*Excoffier & Lischer, 2010*)

**Table 1  Geographical location of *Limacina helicina* samples and compositions of haplotypes.**

| Location | Coordinates/ References | | N | Haplogroup | N from haplogroup | GenBank accession number |
|---|---|---|---|---|---|---|
| | N° | E° | | | | |
| Kara Sea, South St. 5265 | 70°53 | 58°18 | 22 | H1 | 3 | MH379290–MH379311 |
| | | | | H2 | 19 | |
| Kara Sea, Voronina Trough, St. 5239 | 78°36 | 88°04 | 12 | H1 | 2 | MH379312–MH379330 |
| | | | | H2 | 10 | |
| Kara Sea, St. Anna Trough, sT. 5212 | 76°43 | 70°59 | 23 | H1 | 7 | MH379266–MH379289 |
| | | | | H2 | 16 | |
| Svalbard | *Sromek, Lasota & Wolowicz (2015)* | | 68 | H1 | 11 | AB859527–AB859593 |
| | | | | H2 | 57 | |
| Pacific Ocean | *Jennings et al. (2010)*, *Chichvarkhin (2016)*, *Shimizu et al. (2017)* | | 105 | H1 | 105 | FJ876923, KX871888, KX871889, LC185015–LC185073, LC229727–LC229769 |
| | | | | H2 | 0 | |
| Canadian Arctic | *Hunt et al. (2010)*, *Layton, Martel & Hebert (2014)*, *Jennings et al. (2010)* | | 6 | H1 | 6 | GQ861826–GQ861828, HM862494, HM862496, FJ876924 |
| | | | | H2 | 0 | |

**Notes.**

*N*, number of analyzed individuals; H1 and H2, haplogroups.

software was used for pairwise ϕST calculations between regions analysis and verification of neutrality. Significance of ϕST was tested with 1,000 permutations.

## RESULTS

We analyzed 73 specimens of *L. helicina* from the Kara Sea. COI sequences were obtained from 65 of these samples, and eight sequences were discarded due to poor quality contigs, leaving 57 individual sequences for further analysis. The highest ϕST value found between the southern and northern parts was 0.023 (n. s.) (Table 2). Due to the lack of significant differences in genetic structure between the three different Kara Sea collection locations, the data from these stations were combined for comparison with Svalbard and Pacific populations. In total, 179 *L. helicina* sequences from the Arctic and Pacific were downloaded from GenBank (Table 1). These sequences were regarded as three large geographical subgroups: the Kara Sea, Svalbard (data from *Sromek, Lasota & Wolowicz, 2015*), and the Pacific (*Jennings et al., 2010*; *Chichvarkhin, 2016*; *Shimizu et al., 2017*). We also added the data from the Canadian Arctic (*Hunt et al., 2010*; *Jennings et al., 2010*; *Layton, Martel & Hebert, 2014*) for the haplotype network construction. A total of 65 haplotypes were found from all sequences, which were combined in two large haplogroups, which differ from each other by 2 nucleotide substitutions (Fig. 2B). Each haplogroup represents a typical star-like haplonet with numerous branches. These patterns are in agreement with analysis of *Shimizu et al. (2017)* and so we adopted their names of haplogroups as H1 and H2. Haplogroup H2 includes the majority of sequences from the Kara Sea and Svalbard,
**Table 2  Pairwise Phi-st values and associated *p*-values among *L. helicina* populations from the three sampling sites in Kara Sea and three different geographical areas.** Significant differences ($p < 0.001$) are in bold.

| Compared areas | $\Phi$st | *p*-value |
|---|---|---|
| Within Kara Sea | | |
| St Anna–Voronin | 0.01574 | 0.21622 |
| St Anna–South | 0.02292 | 0.15315 |
| Voronin–South | −0.00263 | 0.42342 |
| Between different seas | | |
| Kara Sea–Svalbard | −0.00109 | 0.47748 |
| **Kara Sea–Pacific** | **0.63422** | **0.00000** |
| **Svalbard–Pacific** | **0.60013** | **0.00000** |

while the H1 is widespread at all research locations and found throughout all *L. helicina* populations (Table 1).

The samples from Kara Sea were represented by 19 haplotypes with two being widespread (Fig. 2A). The remaining haplotypes are structured by their variations, differing by 1–3 nucleotide substitutions. The majority of the Kara Sea individuals are represented by the H2 haplogroup (79%). The greatest variability of haplotypes was found at the St. Anna Trough in the north of the sea. The H2 haplogroup was also the predominant haplogroup found in samples from Svalbard fjords (84%), while only H1 haplogroups were found in the Pacific region (Fig. 2B). The highest haplotype diversity (Table 3) was reported around Svalbard ($H = 0.771$). The diversity of haplotypes in the Kara Sea is similar ($H = 0.672$), despite a smaller number of analyzed individuals. The diversity of haplotypes in Pacific is significantly lower ($H = 0.449$) as well as nucleotide diversity (Table 3). The Tajima's D and Fu's Fs neutral evolution model tests showed significant negative values (Table 3).

The haplotype network (Fig. 2B) shows similarity between the Kara Sea and Svalbard populations. The ratio between the H2 and the H1 haplogroups in this region is also similar—the majority of individuals belongs to the H2 haplogroup. All individuals from Canadian Arctic and Pacific share the H1 haplogroup.

Pairwise comparison of φST showed no significant differences between the Kara Sea and the Svalbard populations (φST = −0.00109, n. s.); however, the samples from the Kara Sea and Svalbard differed significantly from the Pacific (Table 2).

## DISCUSSION

The obtained results support our hypothesis that *L. helicina* populations from the Kara Sea would be genetically similar to those near Svalbard.

The haplotype network is very similar for populations from the Kara Sea and those from the North of the Atlantic near Svalbard (Fig. 2B), and the ratio of haplotype group H2 to haplotype group H1 is also similar. The majority of haplotypes belongs to haplotype group H2, a minor part to the group H1. In contrast, in the Canadian Arctic and the Pacific only haplogroup H1 is found. The H1 group of haplotypes is widespread and occurs at all stations and populations (Table 1, Fig. 3), and populations from the Pacific Ocean and

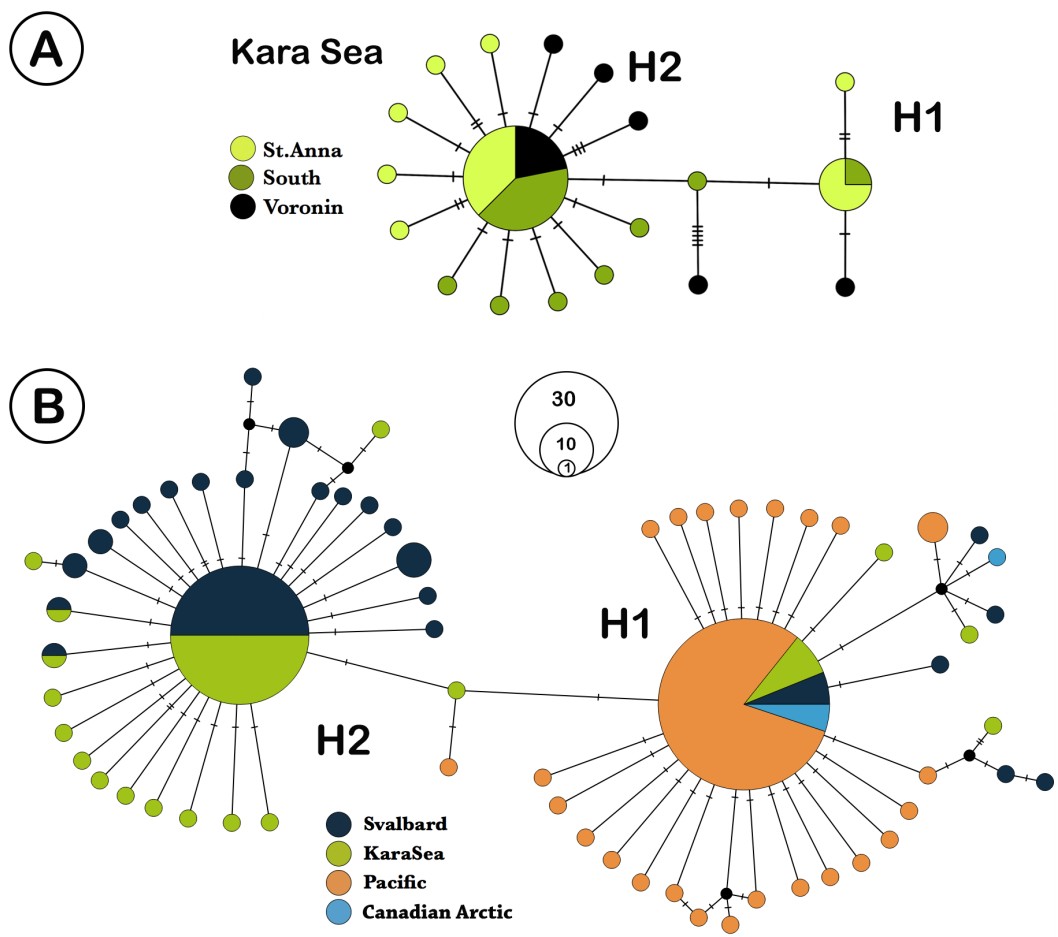

**Figure 2** **TCS network of *Limacina helicina* haplotypes.** (A) haplotypes from Kara Sea (this study). St. Anna trough is marked in light green, Voronin through—in black, and southern part of Kara Sea—in dark green. (B) haplotypes across Northern hemisphere based on the current research and the GenBank data. Svalbard population is marked in dark blue, the Kara Sea in green, Pacific in orange, and the Canadian Arctic in blue. H1—haplogroup 1, H2—haplogroup 2. Notes: each haplotype is colored according to the location where it was collected. Haplotype circle sizes indicate frequency (according to the Table 1).

the Canadian Arctic were almost identical and were represented by the same sequence. This is explained by the main currents through the Bering Strait and indicates the possible direction of distribution of plankton communities from the Pacific Ocean (*Nelson et al., 2009*; *Questel et al., 2016*). Typical star-like haplonet and the conducted Tajima's D and Fu's Fs tests can point to the rapid population expansion. The negative values of Fu's Fs indicate the presence of a large number of low frequency haplotypes, usually described for loci under directed selection or expansion of the population after a severe decline (however, see (*Niwa, Nashida & Yanagimoto, 2016*), for an alternative explanation of negative D and F in abundant marine organisms). The reason for this pattern could be the rapid expansion of this species after the last glacial maximum. Similar dispersal was observed for other Arctic

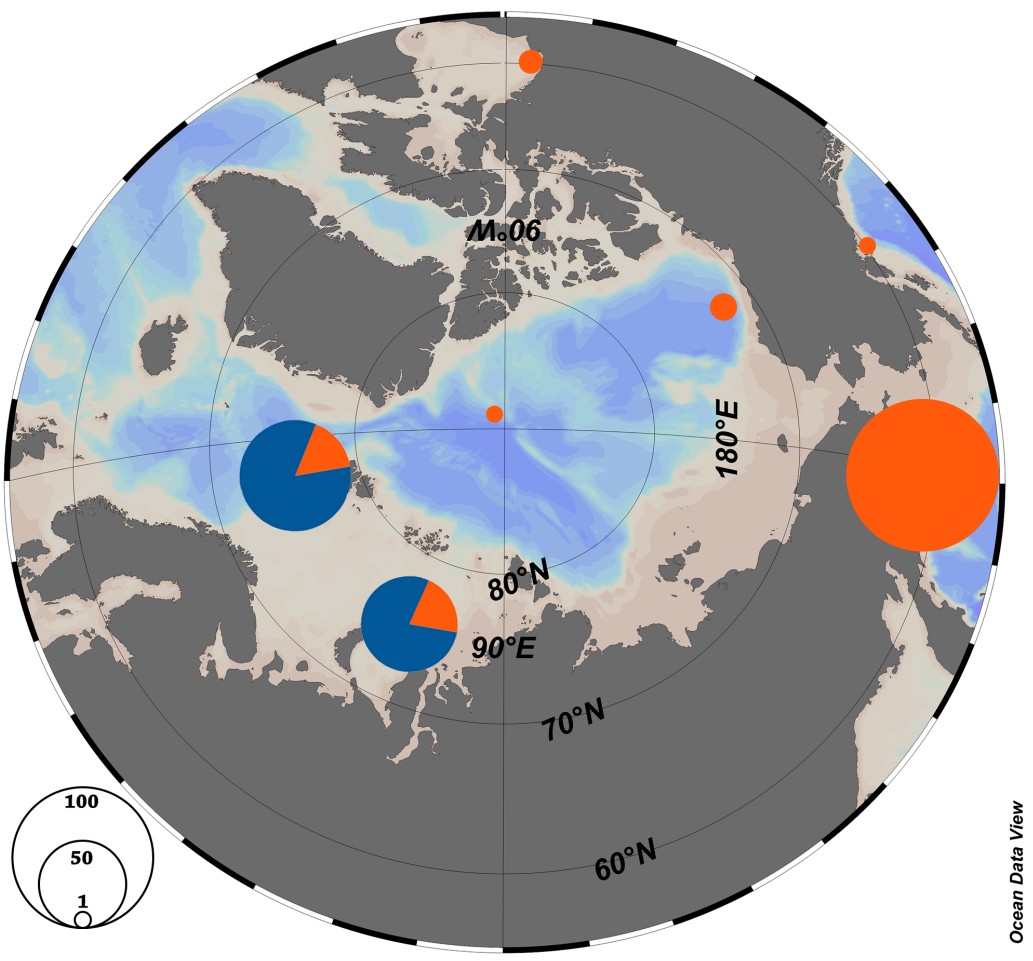

**Figure 3** **Haplotype distribution among the Arctic populations of *Limacina helicina*.** Orange—haplogroup H1, Blue—haplogroup H2. *Schlitzer (2018)*.

**Table 3** **Estimates of genetic diversity in populations of *L. helicina* from regions of Arctic and Pacific Ocean.** Nucleotide and haplotype diversity, neutrality test.

| | N | Ns | k | S | H | $\pi$ | $\Pi$ | D | Fs |
|---|---|---|---|---|---|---|---|---|---|
| Kara Sea | 57 | 500 | 19 | 26 | 0,672 | 0,00301754 | 1,509 | $-2,35896$ ($p < 0.001$) | $-17,725$ ($p < 0.0001$) |
| Svalbard | 68 | 503 | 24 | 25 | 0,771 | 0,00338705 | 1,704 | $-2,10848$ ($p < 0.01$) | $-24,253$ ($p < 0.0001$) |
| Pacific | 105 | 503 | 26 | 26 | 0,449 | 0,00124309 | 0,625 | $-2,60329$ ($p < 0.001$) | $-42,81$ ($p < 0.0001$) |

**Notes.**

N, number of individuals; Ns, number of sites; k, number of haplotypes; S, polymorphic sites; H, haplotype diversity; $\pi$, nucleotide diversity; $\Pi$, average number of nucleotide differences; D, Tajima's D; Fs, Fu's Fs neutrality test.

species that have survived in the refugia, then quickly spread to their current habitats after the deglaciation (*Hewitt, 2000*; *Weydmann et al., 2017*).

According to the previous studies (*Sromek, Lasota & Wolowicz, 2015*; *Shimizu et al., 2017*), *L. helicina* were formerly widely distributed in the Arctic and the Pacific, but the populations were isolated in the Northern Atlantic during the glaciation. Haplotype group

H1 may have persisted in a Pacific refuge, and H2—in an Atlantic refuge. Subsequently, during the retreat of the glacier about 131 ky BP, there was an increase in genetic diversity and distribution around Svalbard (*Sromek, Lasota & Wolowicz, 2015*). A similar spread of Pacific fauna was shown for other groups of organisms in the Atlantic region (*Laakkonen et al., 2013*). The recent distribution of *L. helicina* haplotypes could be explained in a similar way. When the ice sheets disappeared between the Pacific and Atlantic, the Pacific population could have resettled in the Arctic. This hypothesis is supported by the existence of a separated haplotype group H1 along with haplotype group H2 (Fig. 3). The currents flowing between the Pacific and the Arctic through the Bering Strait have a predominantly northward direction (see references in *Questel et al., 2016*). This lends support to a hypothesis that Limacina helicina may only effectively migrate from the Pacific into the Arctic and not the other way around, which is consistent with our observation that the H1 haplogroup has reached the Arctic, while the H2 haplogroup appears to be absent from the Pacific (Fig. 3)

The absence of significant differences between the Kara Sea and Svalbard and the similarity of proportions of different haplotypes in these regions is consistent with an ongoing or recent exchange between these two populations, which coincides with the oceanography in this area (*Stiansen & Filin, 2007*).

Frequency of occurrence of different haplotypes varies between locations of the Kara Sea (near the Kara Strait, the St. Anna and Voronin Troughs), but these differences are not significant. In the south (station AMK 5265 near the Kara Strait), the percentage H1 haplotype (14%) is lower than in the north at St. Anna or Voronin Troughs (st. AMK 5239, 5212) (26%). This is in accordance with the penetration of water of different origins into the sea: in the south–west at station AMK 5265, the warm and salty water of the Barents Sea origin penetrates through the Kara Strait, while the northern part of the Kara Sea is strongly influenced by the Arctic saline and cold water (*Zatsepin et al., 2015*). Since these populations were not significantly different genetically, the different environments are not isolating either population.

## CONCLUSIONS

This study represents the first research on the genetic structure of *L. helicina* in the Kara Sea and makes an important contribution to zooplankton phylogeography by providing data on this large Arctic sea, which is not easily accessible. The comparison of our own data from the Kara Sea with the published data obtained in the Svalbard area, northwest Pacific, and Canadian Arctic, allowed us to conclude that the distribution of haplotypes in the Kara Sea is similar to that in Svalbard. Although no significant differences between habitats within the Kara Sea were found, the proportion of haplotypes H2 was higher near the Kara Strait than in the northern troughs. The analysis of the available data provides insight into the population structure of this pteropod species, indicating possible direction of post-glacial distribution of *L. helicina* in the Arctic. However, many questions regarding the genetics of this mollusk in the Arctic still remain unresolved, and in future studies we hope to better understand how far the western population of *L. helicina* penetrates and how the haplotypes are distributed over other Arctic seas.

## ACKNOWLEDGEMENTS

For the help in the collection of *Limacina helicina* and discussion of this project we would like to thank AV Drits, as well as VV Aleshin and C Gross for extremely helpful comments on this manuscript. We are particularly grateful for the laboratory assistance from the 2017 Invertebrate Zoology bachelor students (Lomonosov Moscow State University). In addition, we gratefully acknowledge the native English speaker Elizabeth Schmidt for constructive comments of the manuscript. We thank Pieternella Luttikhuizen, Katharina Jörger and an anonymous reviewer for important and valuable comments that significantly improved the paper.

### Funding

This research was performed in the framework of the state assignment of FASO Russia (theme No. 0149-2018-0009) and was supported by RFBR grant #16-04-00064 (sample collections and preparation of manuscript) and RSF grant #14-50-00029 (molecular analysis). The funders had no role in study design, data collection and analysis, decision to publish, or preparation of the manuscript.

### Grant Disclosures

The following grant information was disclosed by the authors:
FASO Russia: 0149-2018-0009.
RFBR: #16-04-00064.
RSF: #14-50-00029.

### Competing Interests

The authors declare there are no competing interests.

### Author Contributions

- Galina Anatolievna Abyzova conceived and designed the experiments, performed the experiments, analyzed the data, contributed reagents/materials/analysis tools, prepared figures and/or tables, authored or reviewed drafts of the paper, approved the final draft.
- Mikhail Aleksandrovich Nikitin performed the experiments, analyzed the data, contributed reagents/materials/analysis tools, prepared figures and/or tables, authored or reviewed drafts of the paper, approved the final draft.
- Olga Vladimirovna Popova performed the experiments, prepared figures and/or tables, authored or reviewed drafts of the paper, approved the final draft.
- Anna Fedorovna Pasternak conceived and designed the experiments, contributed reagents/materials/analysis tools, authored or reviewed drafts of the paper, approved the final draft.

### Data Availability

   GenBank: MH379266–MH379330.

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
