# Peer review of "Genetic population structure of the pelagic mollusk Limacina helicina in the Kara Sea"

_PeerJ, doi:10.7717/peerj.5709_

## Round 0.1 · original submission · Major Revisions

The three referees all found your manuscript interesting and a worthwhile extension of previous work. Referee 1 provide many key specific points which I ask you to address.

The English (wording and grammar) is also in need of serious revision. As most non-native English speakers I can recommend that you make contact with a native English-speaker in the same area of research (i.e., a colleague) to help you with that aspect.

Best regards,

Per [Palsboll]

·

Basic reporting

The English is not yet perfect. I have added a list of examples (not exhaustive) to my general comments.

Experimental design

This is a relevant addition to what was already known about an abundant pelagic mollusk. The authors have gathered data in a difficult to reach Arctic Sea and connected the new data to data published earlier.
The interpretation of results can be improved and elaborated; for suggestions see me general comments.

Validity of the findings

There is a small difference in sample size between what is mentioned in the paper and what I found in the data files provided. For details see General Comments.

Additional comments

Major comments
L106-107 you report on Fst values from the AMOVA and not Phi-st which would be more standard. The latter takes into account nucleotide differences between haplotypes. Are you certain you used the haplotype frequencies only? In that case the use of Fst throughout the manuscript is OK. Please add that to the lines 106-107 and provide arguments why you chose to disregard the sequence information. If you did use the sequence information (preferable), you should add that at lines 106-107, and change Fst to Phi-st throughout the manucript.
L109 reports 57 sequences but the data file 'limacina-helicina-kara.txt' contains 65 sequences.
L112, in contrast, does report on 236-179=57 sequences (236 is the number in 'limacina_helicina_kara_pacific_svalbard_236seq.txt' and 179 is the number of sequences used from other sources as reported here in L112)
L137-138 Contrary to what is written here, Table 2 does not indicate which values significantly depart from 0. This should be added.
About Fu's Fs: throughout the manuscript you interpret the significantly negative values as evidence of a bottleneck (e.g., L157), with or without recent expansion (e.g., L41-42, L159). To my knowledge this is not correct. A negative Fu's Fs means that there are more haplotypes in the sample than expected on the basis of the number of pairwise nucleotide differences and the sample size. This happens (among others) after a recent population expansion. A bottleneck is not necessary to cause the low Fs and so you should not say that.
Within Kara Sea. You sampled at three locations in Kara Sea but do not report on (the absence of) structure among these samples. Only L109-110 reports on one comparison, while the other two other pairwise comparisons as well as the overall Fst for the three samples are missing. It seems obvious from the haplotype frequencies that there is no structure but in my opinion you should report on it nonetheless.
L45-48 (and L204) on the basis of what do you conclude that the origin is Pacific and not Atlantic? Just the Fu Fs analyses seems limited evidence (and the interpretation needs to be reconsidered as mentioned above). Your data do not refute a scenario in which the species originates from the Atlantic, colonized the Pacific quite a while ago with no traces of a bottleneck there, while Pleistocene reductions in the Atlantic cause the bottleneck signature here. Also, the fact that both haplogroups are star-like (as you note yourself in L119-120) makes it likely that both have a similar demographic history. Limacina's do have a fossil record which I suggest you dive into to add more credibility to this claim. Burridge et al 2017 (Plos One) may provide some insights into the location of origin of Limacina species and they also reference fossils.
L185-186 'indicate an exchange between these two populations' This is not correct. The observation is consistent with ongoing exchange but it does not indicate it. It is also possible that population subdivision is recent, and an absence of exchange would not yet show in the haplotype frequencies. I suggest to change 'indicate' to 'is consistent with' and mention the alternative explanation as well.
L192-195 and L200-203 continuing from the previous point, here you contradict yourself. You say that different water bodies do not fully mix in the Kara Sea and that this may underlie (non-significant) genetic differences within Kara Sea. If you just concluded that there is exchange between Kara and Svalbard, but (some) isolation within Kara, that sounds contradicting to me. You need more data to have more certainty.

Minor comments
L27 you analyzed 65 haplotypes of which 19 Kara Sea. Better would be to report how many new individuals you sequenced (Kara Sea), how many haplotypes were among them, and how this compares to the number of haplotypes found previously by other authors.
L31-32 'While the other one, named Pacific, is widespread and found throughout all L. helicina populations.' Also at L122-123. It seems strange to choose a regional name to describe a haplotype group that is found everywhere. I suggest to use a different name.
L32-35 the description of the structure that you found is not precise. Please change it so that the reader knows exactly at what level you did detect structure and at which level you did not.
L45-46 what do you mean by 'the form' here? (same at L166-168)
L157 'by populations between the Pacific and the Arctic' it is unclear to me what you mean here
L159-160 'The observed wide diversity of haplotypes indicates a large population size' also not clear what is meant here
L179 'an isolated Atlantic haplotype along with the Pacific haplotype (Fig. 3)' I do not see what exactly in Fig. 3 you mean here
L179-181 'The main northward currents from the Pacific to the Arctic region are nowadays passing through the Bering Strait' I suppose you mean the main northward currents going into the Arctic, because what other routes into the Arctic from the Pacific are there except the Bering Strait?

Spelling/grammar etc.
L20 delete 'the data'
L32-33 sentence is not grammatically correct
L39 with 'samples' do you mean 'numbers'? (same for L154)
L39-41 is a repetition from what you write just before it (same for L153-155)
L41 delete 'the' in 'indicate the directed selection'
L42 change 'the passage of the bottleneck' to 'a population bottleneck'
L114 delete 'the' from 'the Svalbard'
L132 change 'Atlantic haplogroup' to 'The Atlantic haplogroup'.
L143-144 consider to rephrase 'suggestion about genetic similarity of L. helicina populations from the Kara Sea and around Svalbard' to something like 'expectation that L. helicina from the Kara Sea would be genetically similar to those near Svalbard'.
L147 delete 'the' in 'to the haplotype group'
L148 delete 'is' in 'minor part is to'
L158 I do not think 'redundancy' is the right word here
Table 2 and elsewhere: decimal in English is done with a period not a comma (so 1.5 not 1,5)
Figure 3 caption: please provide more information, e.g. size of circle indicates sample size.
L192 change 'Tis' to 'This' and 'according' to 'accordance'

Reviewer 2 ·

Basic reporting

The manuscript is pretty well written (but I am not a native speaker either).

References are fine.

Good structure.

Well presented research, only one "problem":

I consider (very) confusing the fact that authors chose to name the two haplotype clades as Atlantic and Pacific, and the also the regions Atlantic and Pacific. I can follow it (since I am familiar with haplotype networks), but I would recommend maybe just calling them I and II, or something like that. Especially when you have Pacific haplotypes in the Atlantic sample, but not Atlantic haplotypes in the Pacific sample. This phrase, as you can read, can be very confusing. If you name the two clades I and II, then you have no problems: “haplotypes from clades I and II were present in the Atlantic, meanwhile only Clade I ones were present in the Pacific”. Much easier to understand.

Experimental design

Few comments regarding methods:

Did not see any ANOVA done. There are sample by sample Fst distances, not AMOVA. So, methods have to be changed to that, like” Between samples Fst distances were calculates in Arlequin…”

Some results are not shown – distances between the samples from the authors. “We obtained 57 sequences from the Kara Sea. The highest Fst value found between the southern and northern parts was 0.023 (p < 0.5)” but they have three samples. They can add there a “data not shown”

In my opinion, using only one marker for testing for Tajima and Fu’s is not appropriate/informative for planktonic species with such hugh population sizes. They ALL show negative values. See Wares 2010 Evolution 64: 1136-1142.

Validity of the findings

Nice paper regarding population structure in the Arctic, a hot topic nowadays due to climate change.

data is robust and well analyzed.

·

Basic reporting

The article is written in unambiguous, professional English, but it might benefit from last correction by a native speaker. I am not a native speaker myself but I would e.g. consider a definite article necessary in the title: Genetic population structure of the pelagic mollusk…
All relevant literature is cited and provides sufficient background for the understanding of the manuscript. However, the formatting of the references contains several mistakes and needs to be revised carefully. E.g. most of the species names in the titles are not in italic (see number 3,13,16,23 etc.). Sometimes the Journal names are misspelled (i.e. capital letters missing in 16, 27…) Identical articles appear twice (e.g. 16 and 17, 26 and 27) and/ or lack list of authors, citation 35 is incomplete.

Experimental design

No comment.

Validity of the findings

No comment.

Additional comments

The manuscript entitled „Genetic population structure of pelagic mollusk Limacina helicina on the Kara Sea“ (# 23753) by Abyzova et al. presents novel genetic data on the population structure of an ecologically important pelagic gastropod form the Siberian Arctic Sea. It fills (part of) the gap in population genetic data between previous studies from the Svalbard area and from the subarctic western North Pacific, presenting novel insights into haplotype diversity and connectivity. Limacina helicina plays a key role in studies on ocean acidification and climate change, thus I believe that the presented data is of general interest and presents a valuable contribution in documenting current population patters. I clearly want to recommend the manuscript for publication, but would like the authors to consider and review the following questions/ suggestions below prior to publication:
One question: is it correct to speak of different “habitats” (see Abstract and Line 79 and further on) in regards to your sampling sites? In my understanding a habitat provides specific abiotic or biotic factors. Which factors/ parameters differ between your sampling sites that justifies calling them different habitats? In the Conclusions you refer to different “biotopes” – please explain in more detail how the three localities differ.
I would recommend rearranging your results: why not present first the analyses on your Kara Sea samples alone (maybe also adding an additional haplotype network including only Kara Sea specimens)? It could easily be included in Fig. 2 as A and B and would offer the opportunity to show the differences between your three sampling localities and how the haplotypes are spread among them. Once you presented your own data, you could then continue by analyzing your data in the context of the published sequences from the different geographic areas.
Minor corrections in the text:
Line 53: locally
Line 55: there is no Hunt 2008 in the references, should probably read Hunt et al. 2008
Line 76: Limacina in italics
Line 96: Out of curiosity: how was the percentage of successful COI sequences? I.e. how many specimens were available to the study and how many did you sample to get your sequences?
Line 106: Excoifer
Line 112: In total, 179 L. helicina …
Line 117: Layton et al., 2017
Line 143: maybe replace “suggestion” with “hypothesis”
Line 168/170: Sromek et al, 2015
Line 174: A similar spread…
Line 175: Laakkonen et al. 2013
Line 177: during the glacial period between…
Line 192: This is in accordance with
In the tables you refer to Spitsbergen – which is only called Svalbard throughout the main text.

My apologies for the delayed review!

---

## Round 0.2 · Minor Revisions

The referees were happy with your revisions. They both suggest remaining minor changes which I suggest you undertake before we accept the manuscript.

Best regards,

Per [Palsboll]

·

Basic reporting

1. The authors did an overall good job of incorporating the reviewers' comments. I have a number of additional suggestions, but they are mostly minor.
2. L17-20 these two sentences make it sound as if two different data sets were collected to test for a) difference among locations, and b) difference among habitats. But they are the same comparison among the same three samples, aren't they? So this should be rephrased to reflect that.
3. L28 'combined into two significantly different groups'. The word 'significantly' here is not based on a statistical test and should be omitted.
4. L62 do you really mean per square meter here, or per cubic meter instead? It is a pelagic organism after all, so measuring their density in 2D seems unlogical.
5. L66 you mention here 'Limacina' but because you mean the species here I suggest to change it to 'Limacina helicina' (in italics).
6. L79-81 The salinities lack a unit.
7. L106-109 It is not common practice to write p < 0.5. Instead, I suggest to write n.s. for not significant. The same for L137.
8. L106-109 please refer here to your Table 3.
9. L168 H1 and H2 are haplogroups (or haplotype groups) but not haplotypes

Experimental design

10. L162-164 Not clear if this sentence refers to your own data or analyses, or if it refers to a general pattern for other taxa. Please make this more clear. If it refers to a general pattern, please provide references. If it refers to your own data/analyses, I am not convinced that this can be concluded; you did not do analyses that show that diversity increased at that time.
11. L168-172 This is still not explicit enough after revision. I would suggest to change 'The northward currents from the Pacific to the Arctic region are currently passing through the Bering Strait (Palumbi & Kessing, 1991; Questel et al., 2016), which is reflected by the wide distribution of the haplotypes H1 throughout the Arctic and by the absence of haplogroup H2 south of the Bering Strait in the Pacific Ocean.' to: 'The currents flowing between the Pacific and the Arctic through the Bering Strait have a predominantly northward direction (see references in Questel et al., 2016). This lends support to a hypothesis that Limacina helicina may only effectively migrate from the Pacific into the Arctic and not the other way around, which is consistent with our observation that the H1 haplogroup has reached the Arctic, while the H2 haplogroup appears to be absent from the Pacific (Fig. 3)'.
12. L169-170 Palumbi & Kessing, 1991 is not a good reference here.
13. L187-188 Indeed, this paper makes an important contribution to zooplankton phylogeography by providing data on this large Arctic sea, which is not easily accessible.

Validity of the findings

14. all is good

Additional comments

Grammar and spelling:
15. L27 change 'around Arctic' to 'around the Arctic'
16. L38 delete 'due to'
17. L53 change 'for' to 'studies on'
18. L64 change 'shelf Siberian' to 'shelf for the Siberian'
19. L79 change 'this station affected' to 'this station is affected'
20. L119 change 'includes majority of' to 'includes the majority of'
21. L125 change 'haplogroup also' to 'haplogroup was also'
22. L127-129 I am not a native speaker but I think it is better to use 'diversity' instead of 'variety'
23. L143 change 'North of Atlantic' to 'North of the Atlantic'
24. L154 change 'for alternative explanation' to 'for an alternative explanation'
25. L162 change 'in Pacific refuge' to 'in a Pacific refuge' and 'H2 - in Atlantic' to 'H2 in an Atlantic refuge'.

Reviewer 2 ·

Basic reporting

no comments

Experimental design

Again – there was no AMOVA done. I do not see any analysis of the partitioning of the variance. The only think I see there is pairwise ϕST distances between samples or regions, not an AMOVA. Methods should then read “Pairwise ϕST distances were calculated in Arlequin between samples within the Kara Sea and between seas”. That IS NOT an AMOVA analyses. An AMOVA will give you three statistics (ST, SC, CT), about how the genetic variance is distributed within samples, between samples within groups and between groups. It is true that they the same screen in Arlequin, but... Results explain properly what the authors did though.

Validity of the findings

no comments (after previous review)

Additional comments

Lines 114 and 115: “A total of 65 haplotypes was found” should read “A total of 65 haplotypes were found”. The noun is still “haplotypes”. Only singular if “total” is the noun.

Substitute “Phi-st” by ϕST (the greek letter) in the text

---

## Round 0.3 · accepted · Accept

Thank You for undertaking the revisions. The manuscript is now acceptable.

Thank You for submitting your work to PeerJ

Cheers,

Per

# ·

Basic reporting

The authors did a good job of incorporating reviewer comments. I have no further issues.

Experimental design

No further issues.

Validity of the findings

No further issues.

Reviewer 2 ·

Basic reporting

All my questions have been answered / solved

Experimental design

All my questions have been answered / solved

Validity of the findings

No comment

Additional comments

All my questions have been answered / solved